# Alterations in Serum Concentration of Soluble CD163 within Five Study Days from ICU Admission Are Associated with In-Hospital Mortality of Septic Patients—A Preliminary Study

**DOI:** 10.3390/ijerph20032263

**Published:** 2023-01-27

**Authors:** Magdalena Mierzchała-Pasierb, Małgorzata Lipińska-Gediga, Łukasz Lewandowski, Małgorzata Krzystek-Korpacka

**Affiliations:** 1Department of Biochemistry and Immunochemistry, Wroclaw Medical University, 50-368 Wroclaw, Poland; 2Department of Anesthesiology and Intensive Therapy, Wroclaw Medical University, 50-556 Wroclaw, Poland

**Keywords:** sCD163, interleukin 18, sepsis, septic shock, biomarkers

## Abstract

Background: CD163, a cell membrane surface molecule specifically expressed by macrophages with an anti-inflammatory phenotype, participates in innate immunity. The purpose of the study was to evaluate the clinical utility of sCD163 in septic patients in comparison to other parameters associated with infections, mainly PCT, CRP and IL-18. Methods: Serum samples were obtained from 40 septic patients on the ICU admission day, 3rd and 5th study days. The control group consisted of 30 healthy volunteers from whom the specimen was collected once. An enzyme-linked immunosorbent assay (ELISA) was used to determine the concentrations of sCD163 and IL-18. CRP and PCT records, among others, were provided by the hospital. Results: Septic shock was associated with the highest concentrations of sCD163 and IL-18. Admission values of sCD163 significantly contributed to mortality prediction in septic patients. Conclusions: The concentration of sCD163 determined on the ICU admission day may potentially be utilized in estimation of the odds of death among septic patients.

## 1. Introduction

Sepsis and its associated complications, being the main factors for high morbidity and mortality in intensive care units (ICU) [1], stem from the dysregulated pathogen-driven response of the immune system. In physiological conditions, molecular response is mediated by the activation of macrophages, followed by an expression of many surface receptors and a release of various inflammatory mediators [2] such as the CD163 receptor (hemoglobin (Hb) scavenger receptor) and interleukin (IL)-18.

CD163, a surface glycoprotein and a member of the group B scavenger receptor cysteine-rich family, functions as a receptor for hemoglobin-haptoglobin (Hb-Hp) complexes. It is superficially expressed on tissue macrophages, blood monocytes and lymphocytes (B and T). In humans, CD163 consists of a large extracellular region, 9 cysteine-rich class B domains, a transmembrane fragment and a C-terminal short cytoplasmic tail [3]. CD163 occurs in two forms: the soluble form (sCD163, found in the serum/plasma), and the cell membrane-associated form (mCD163). The sCD163 is formed through separation of the extracellular domain from mCD163 by activating agents that affect monocytes or macrophages [4]. The macrophage-associated CD163 receptor is responsible for the uptake and endocytosis of Hb or Hb-Hp complex [5]. The expression of the sCD163 receptor is controlled by tumor necrosis factor-α (TNF-α) and IL-10 [6,7].

Activation of monocytes by bacterial lipopolysaccharide contributes to an increase in plasma concentration of sCD163 [8]. Various structural components of bacteria and yeasts cleave mCD163, leading to liberation of sCD163 from the surface of monocytes. Upon its release, sCD163 binds free Hb and Hb-Hp complexes, depleting the source of iron for the pathogenic bacteria. Interestingly, sCD163 is also involved in the late phase of an inflammatory response [9]. Its interaction with Hb-Hp complexes leads to a release of strong anti-inflammatory agents—IL-10 and carbon monoxide [10].

It is indicated in the literature that some bacterial infections are associated with high levels of the sCD163 receptor [11,12]. Sulahian et al. hypothesized that the surface expression of CD163 on monocytes is regulated during an early immune response [4]. Hence, the cleavage of the extracellular domain of CD163 occurs through the activation of the surface Toll-like receptors [13]. Thus, sCD163 may potentially act as an important component of the innate immune response.

Due to its structural and functional similarity to IL-1, IL-18 is classified in the IL-1 family of cytokines [14]. Macrophages and dendritic cells are the main sources of the active form of IL-18. Intracellularly, IL-18 remains as an inactive precursor in mesenchymal cells [15]. Increased expression of both inactive and active forms of IL-18 was observed in various cell types: T and B lymphocytes, keratinocytes, adrenal cortex cells, chondrocytes, epithelial cells and Browicz-Kupffer cells [16]. Similar to IL-1α or IL-33, IL-18 is released from apoptotic cells, (presumably) owing to the activity of the enzymes neutrophil protease and proteinase-3 [17].

The IL-18 receptor is a heterodimer composed of the subunit IL-18Rα and IL-18Rβ. The α subunit is responsible for extracellular cytokine binding, while the β chain is responsible for signal transduction [18]. The activity of IL-18 is regulated by a soluble receptor IL-18-binding protein, which exerts high affinity towards IL-18, preventing its (IL-18) interaction with the receptors of the immune system-associated cells [14].

IL-18 plays a key role in maintaining the homeostasis between the Th1- and Th2-associated cellular responses. Both IL-18 and IL-1 utilize the same pathway to activate NF-κB and induce the release of chemokines, adhesive molecules and the Fas ligand [19]. Infections, autoimmune and neoplastic diseases have been associated with an elevated concentration of IL-18 [20].

Given the abovementioned considerations, we decided to classify the IL-18 as representing the pro-inflammatory and sCD163 as representing the anti-inflammatory derivatives of sepsis/septic shock.

The purpose of this study was to investigate whether serum concentrations of sCD163 change over time among patients with sepsis and septic shock and to determine its prognostic significance in septic patients. Serum concentrations of IL-18, procalcitonin (PCT), C-reactive protein (CRP) and other biochemical, hematological and clinical parameters acted as a reference in this part of the study.

## 2. Materials and Methods

### 2.1. Study Population

Our study was observational and prospective where the study population comprised 40 septic patients (16 patients with sepsis and 24 patients with septic shock). The control group consisted of 30 healthy volunteers. The septic patients were admitted to the Department of Anesthesiology and Intensive Care of Wroclaw Medical University, Wroclaw, Poland between the period November 2016 and November 2017 and were enrolled in the study upon meeting the criteria of sepsis or septic shock. Sepsis and septic shock were classified according to the new definitions (Sepsis-3) [1]. According to these criteria, sepsis is defined as a life-threatening organ dysfunction caused by a dysregulated host response to infection. The clinical criteria for inclusion were suspected or documented infection and an acute (≥ 2) increase in the SOFA (sequential organ failure assessment) score, which is based on organ dysfunction assessment. Septic shock is defined as a subset of sepsis in which the underlying circulatory, cellular and metabolic abnormalities are profound enough to substantially increase mortality. The inclusion criteria for classification of septic shock were sepsis and vasopressor therapy needed to elevate mean arterial pressure ≥65 mm Hg and lactate >2 mmol/L (18 mg/dL) despite an adequate resuscitation volume [1]. The exclusion criteria were age below 18 years, pregnancy, immunosuppressive treatment and terminal illness with no chance of meaningful recovery, cancer (suspected or confirmed), ongoing chemotherapy and patient death or discharge within 72 h after ICU admission. All patients were treated according to the recommendations listed in the Surviving Sepsis Campaign guidelines [1].

Organ dysfunction was assessed with the SOFA score [21], during a five-day follow-up. Clinical condition severity was assessed with use of the APACHE II (acute physiology and chronic health evaluation II) score [22] upon admission to the ICU. Patient status was tracked until hospital discharge (survivors) or death (non-survivors). In the whole cohort, the mortality rate was 27.5%. Fatal outcome was observed among 45.8% of patients suffering from septic shock.

Healthy volunteers were recruited from the staff of the Diagnostics Laboratory for Teaching and Research of the Wroclaw Medical University. The median age of the participants was 61 years (ranged from 52 to 84 years).

### 2.2. Ethical Considerations

The study protocol was approved by the Medical Ethics Committee of Wroclaw Medical University, Wroclaw, Poland and was conducted in accordance with the Helsinki Declaration of 1975, as revised in 2008. The patients (or their legal representatives) and healthy volunteers signed an informed consent before enrollment in the study.

### 2.3. Laboratory Analysis

Blood samples were collected into serum-separator tubes at three time points within 24 h after ICU admission (1st day), and on the 3rd and 5th days after admission. After specimen collection, tubes with blood were left until a blood clot was formed (15 min, room temperature). The serum was centrifuged (15 min, 720 g), aliquoted and stored at −80 °C until the analysis.

An enzyme-linked immunosorbent assay (ELISA) was performed to measure serum concentrations of IL-18 and sCD163. The tests were performed in accordance with the protocols provided by the manufacturer (Human IL-18 Platinum ELISA, eBiosciences Dx Diagnostics, Vienna, Austria and Human CD163 Immunoassay, R&D Systems, Minneapolis, MN, USA, respectively). The sample absorbance was measured at 450 nm in an ELISA microtiter plate reader (Infinite M200 Tecan, Grödig, Austria). All measurements were performed in duplicate so as to increase precision.

PCT, CRP, white blood cell count (WBC), red blood cell count (RBC), blood platelet count (PLT) and concentration of creatinine were determined by an accredited, in-hospital laboratory. Clinical and demographic data were collected upon specimen collection.

### 2.4. Statistical Analysis

Statistical analysis was performed with use of STATISTICA 13.3, on license of Wroclaw Medical University. The Shapiro-Wilk test was used to check the assumptions of distribution normality. In case of analyzing the differences between dependent samples, Mauchly’s test was used to assess sphericity. Depending on whether the data followed the assumptions of the used statistical methods, repeated measures of ANOVA or the Friedman test were used in the analysis of dependent (time-related) samples and the Kruskal-Wallis ANOVA was used in the analysis of independent (group-related) samples. Given that there was no sphericity, MANOVA (Pillai’s trace) was used. Post-hoc analysis was performed with the use of Tukey’s HSD tests (parametric tests) or according to Siegel-Castellan (non-parametric tests) [23].

In this study, it was the AI which proposed the optimal set of parameters (for performing logistic regression) out of many possible combinations—to remove potential, human-associated bias. However, the estimated odds presented in this study are prone to imprecision stemming from the small population sample size. Based on the sample size, cross-validation has been chosen as the method used to test the analyzed logistic regression models and provide their preliminary metrics. The metrics presented in this study were used solely to compare the models—by no means should they be deemed precise. This imprecision is associated not only with the mentioned small sample size, but also with the fact that better testing algorithms than the one used in this study (10-fold cross-validation) exist. Leave-one-out cross-validation (LOOCV) could be proposed as such an algorithm since its mean accuracy remains constant. Conversely, in K-fold cross-validation, the accuracy bias depends on the number of folds and instances within them, among others [24].

To assess the association between the selected variables and the odds of in-hospital mortality, multiple logistic regression was used. The ***p***-values of the LR test are given in Table A1. The Box-Tidwell test was used to check whether the continuous variables vs. the logarithm of odds dependence was linear. If this assumption was not met, the variables were transformed and checked again for linearity. As the raw values of the sCD163 concentration did not meet this requirement (Table A2, *p* = 0.022), the variable was transformed. Out of two common transformations, the log1.1(CD163) was used, as it was linear vs. log (odds). The Hosmer-Lemeshow test was used to check whether the observed event rates did not significantly differ from the expected event rates. K-fold cross validation (K = 10) was utilized in the analysis of the predictive power of the featured multiple logistic regression model. Continuous variables were preferred over categorical, if feasible. The iterative process of creating the multivariate model of the best fit is shown in Table A3. As it can be observed based in this table, two separate approaches were followed. In one of them, sCD163 and IL-18 concentration were used in the process of iteration. In the other approach, both variables were ruled out of the initial set of features and substituted with their ratio (sCD163/IL-18). The models of the best fit were obtained by the means of backward elimination (*p* cutoff: 0.05). Due to low the sample size, interactions were not included in the analysis. Cut-off points for this model were analyzed with the use of Youden’s J statistic (Table A4).

Survival analysis was performed with a Cox proportional hazards regression. The analysis was carried out in the following process: choosing candidate variables, which could affect the risk of death, reducing the dimensionality (number of variables) of the initial model and analyzing the model prediction metrics to compare each model. Continuous variables were preferred over categorical. Only the variables which featured the values associated with the ICU admission (1st day) were used. Information on a multivariate model consisting of all the initially-chosen variables is shown in Table A5. Results from the likelihood ratio (LR) type 3 test are given in Table A6. Based on iteration (stepwise regression with a 0.05 *p*-value for inclusion and a 0.10 *p*-value for exclusion from the model), two prediction models were selected. Each separate variable featured by the models of the best fit was checked in terms of satisfying the assumptions of the Cox model, based on Schoenfeld residual analysis (Table A7).

Data pre-processing and visualization were performed with use of Python 3.9.13. The following Python packages were utilized: numpy 1.23.0, pandas 1.4.3, zepid 0.9.1, matplotlib 3.6.0, seaborn 0.11.2.

## 3. Results

### 3.1. Characteristics of the Studied Population

Detailed characteristics of the study population are shown in Table 1. Upon admission, the septic cohort was divided into two subgroups: sepsis and septic shock.

### 3.2. The Values of sCD163 and IL-18 in the Control Group and Septic Patients on the 1st Study Day

Upon admission, two subgroups (sepsis and septic shock) were characterized by markedly increased concentrations of IL-18 and sCD163. The differences in the IL-18 concentration between sepsis and septic shock and in the sCD163 concentration between the control group and sepsis were insignificant (*p* = 0.3340; *p* = 0.2522, respectively). However, there was a trend in the median values between the three groups in the concentrations of IL-18 and sCD163: the lowest values were observed in the control group, the intermediate values in sepsis, and the highest values in septic shock.

Interestingly, the sCD163/IL-18 ratio was not subject to the aforementioned trend, as its values were markedly higher in the control group compared to both septic groups (*p* < 0.0001). Moreover, there was no significant difference in the sCD163/IL-18 ratio between the two septic groups (*p* = 1.000). The descriptive statistics and *p*-values are shown in Table 2.

### 3.3. Values of SOFA, Creatinine, WBC, PCT, CRP, IL-18 and sCD163 in Sepsis during the Five-Day Follow-Up

Out of the analyzed parameters (Table 3A), only SOFA score and serum concentrations of creatinine, PCT and CRP revealed significant changes within five days after ICU admission in sepsis. The SOFA score and creatinine concentration significantly and steadily decreased over time (*p* = 0.0010; *p* = 0.0310, respectively), although the change in their values could be reported as significant between the 1st and 5th day only (*p* = 0.0007; *p* = 0.0237, respectively). Both PCT and CRP concentrations decreased over time (*p* = 0.0010; *p* = 0.0334). Although no significance in the PCT concentration was found between the 1st and 5th day (*p* = 0.1000), the difference between the 3rd and 5th day was highly significant (*p* < 0.010). The change in the CRP concentration was significant between the 1st and 5th day (*p* = 0.0024) and 3rd and 5th days (*p* = 0.0186). Interestingly, a decreasing trend in the IL-18 concentration was seen, although the ***p***-value for global differences was on the borderline of statistical significance (*p* = 0.0874). Perhaps, this value was inflated by an increase in variance over time (lowest upon admission, highest on the 5th day). The factors which might affect this change in variance could be studied in future research.

### 3.4. Values of SOFA, Creatinine, WBC, PCT, CRP, IL-18 and sCD163 in Septic Shock during the Five-Day Follow-Up

In septic shock (Table 3B), changes in the values of creatinine, PCT and CRP concentrations followed a similar trend as observed in case of sepsis (Table 3A). Moreover, there were borderline differences in IL-18 concentration (*p* = 0.0787), although the variability decreased over time. The SOFA score did not differ over time (*p* = 0.8698).

Although the observed mean values of the sCD163/IL-18 ratio increased over time, so did the variability of the observed values of this ratio—rendering the time-related differences in this ratio insignificant, regardless of septic shock occurrence (*p*-values were approximately 0.2321 and 0.1088 for sepsis and septic shock, respectively). 

### 3.5. Logistic Regression Analysis

Univariate odds ratios for each parameter, initially used in derivation of the multivariate model, are shown in Figure 1. Only sCD163 was significantly associated with the odds of death (*p* = 0.0299). Each 1.1-fold increase in serum concentration of sCD163 increased the odds of death by approximately 18%.

The multivariate models of the best fit obtained with two iterative approaches are shown in Table 4. The first model (Table 4: model 1) utilized transformed values of sCD163 (log_1.1_) and age. The model was of very good predictive power—accurately predicting approximately 79.2% (testing AUC: 0.792 ± 0.0709) of the survival statuses in the testing datasets during the 10-fold cross-validation. According to the model, each one-year increase in age increased the odds of death by 9.1% (*p* = 0.029). Moreover, each 1.1-fold increase in serum concentration of sCD163 was associated with an increase in these odds by 31.9% (*p* = 0.008).

The second model (Table 4: model 2) utilized three features: the sCD163/IL-18 ratio, concentration of CRP and values of the APACHE II score. The predictive power of this model was inferior to model 1 (model 2 testing AUC: 0.792 ± 0.0709), although different insights could be gathered with its use. According to this model, each one-unit increase in the sCD163/IL-18 ratio was associated with approximately a 3.90-fold increase in the odds of death (*p* ≈ 0.048). Moreover, one-unit increases in values of CRP or APACHE II would increase the odds of death by 0.90% (*p* ≈ 0.026) or 19.0% (*p* ≈ 0.024), respectively.

According to the receiver operating characteristic (ROC) analysis (Table A4), the most optimal P(Y) cut-off value for model 1 (Table 4: model 1) was 0.124, yielding 100% sensitivity and 55.2% specificity. The most optimal P(Y) cut-off value for model 2 (Table 4: model 2) was 0.217; sensitivity and specificity were 0.909 and 0.655, respectively.

Interestingly, not only the values of the sCD163 concentration (as shown in the logistic regression model, Table 4), but also the change of this concentration over the five days from ICU admission (Table 5, Figure 2) are probably associated with in-hospital mortality. A significant difference between non-survivors and survivors in sCD163 concentration was observed during each of three analyzed timepoints (*p*-values approximately 0.017, 0.0005 and 0.0001 for timepoints 1st day, 3rd day and 5th day, respectively). A five-day decrease in the sCD163 concentration was observed in survivors, whereas most of the patients who were about to die showed an increase in these values. Such an association was not inferred in case of IL-18 concentration values.

### 3.6. Cox Proportional Hazards Regression Models

According to the multivariate model of the best fit (Table 6: model 1), every one-unit increase in APACHE II upon admission increased the probability of death by 21.89%. Each consecutive one-unit increase in concentrations of CRP (given in mg/L) and sCD163 (given in ng/mL) increased the hazard rate by 0.73% and 0.18%, respectively. Conversely, an increase in concentration of IL-18 decreased this rate by 0.09% for every 1 pg/mL of IL-18.

Interestingly, the removal of either sCD163 or IL-18 from the aforementioned model greatly affected the prediction of times of death in patients (R^2^ of 0.7600 vs. 0.4831 in model 1 vs. model 2, respectively) and rendered the other variables insignificant in terms of hazard prediction (Table 6: model 2).

## 4. Discussion

Formerly, sepsis was believed to be associated with hyper-inflammatory and anti-inflammatory responses, sequentially following each other [25]. According to the current state of knowledge, these two types of response are viewed as overlapping each other [1]. Regardless of underlying comorbidities and primary injuries responsible for ICU admission, most of critically ill patients are considered to be immunocompromised [26].

In this study, similarly to Kjærgaard et al.’s results [27], IL-18 was significantly elevated in both sepsis and septic shock upon admission, compared to the control group.

IL-18 is a pro-inflammatory cytokine that is produced mainly by antigen presenting cells; it activates NF-κβ inducing inflammatory mediators, especially the interferon (IFN)-γ. In this study, both sepsis and septic shock were characterized by a markedly elevated concentration of IL-18 compared to the control group.

In an animal model, Weiss et al. and Girard-Guyonvarch et al. observed a causative link between IL-18 and the induction of the macrophage activation syndrome (MAS), noting that high levels of free IL-18 (unbound to IL-18BP) increased the risk of developing MAS [28,29]. According to Kyriazopoulou et al., elevated IL-18 levels reflect immune overreaction of the host in sepsis or septic shock. Increased production of IL-1β was associated with a high level of IL-18 [30]. IL-18 induces IFN-γ release resulting in hemophagocytosis being the hallmark of MAS [31].

A slight increase in the concentration of sCD163 was observed in patients with cardiovascular failure [32], chronic renal failure [33], tuberculosis [34], infection [35] or hematological diseases [36].

Our findings of increased sCD163 levels in septic patients agree with previous studies that reported a 1.6-fold higher concentration of sCD163 in critically ill patients compared to healthy controls upon admission [37].

The alternatively-activated macrophages (M2), exhibiting an increased expression of the mannose receptor (CD206) and CD163, differed from the classically-activated macrophages (M1) by the low production of inflammatory cytokines and nitric oxide together with increased production of the extracellular matrix and repair activity [38,39].

In sepsis, the CD163 receptor plays the role of a surrogate marker of monocyte and macrophage modulation and is linked to the anti-inflammatory characteristics of a septic response [40]. Santos et al. observed higher levels of inflammatory cytokines in CD163-positive cells compared to CD163-negative, concluding that these observations were in line with the concept of a dual modulating role of CD163-positive monocytes in the cytokine release in sepsis. This incidence was observed in a model of lipopolysaccharide tolerance, regardless of CD163 expression on the surface of the monocytes [38,40].

Salomaȏ et al. showed that the modulation of the monocyte and macrophage response impaired the release of inflammatory cytokines by these cells, and the efficiency of T cell activation in sepsis. Nevertheless, these cells retained their phagocytic capacity, eradicating microorganisms through generation of reactive oxygen species and nitric oxide [41]. Both Santos et al. and Salomaȏ et al. [40,41] pointed out that this modulation represents a state in which the host organism attempts to stabilize the inflammatory response while maintaining control over the infection. Due to failure to regain homeostasis and clinical recovery, the modulation may result in increased mortality because of the ongoing state of immunosuppression. This observation corroborates with our results, in which a 10% increase in the first-day values of the sCD163 concentration was associated with a 31.9% increase in mortality. These results were confirmed by Cox regression analysis in which a one-unit increase in concentration of CRP and sCD163 increased the hazard rate of death by 0.73% and 0.18%, respectively. Conversely, an increase in concentration of IL-18 decreased this rate by 0.09%. In reference to the ROC analysis, the most optimal cut-off value of the best model (Table 4: model 1) was 0.124, yielding 100% sensitivity and 55.2% specificity. Thus, while not being perfectly accurate, the model may potentially be utilized as an auxiliary, screening test for increased odds of death among septic patients (given that it would positively undergo numerous tests on variable random cohorts).

There was a trend in the median values between the three groups in the concentrations of IL-18 and sCD163: the lowest values were observed in the control group, the intermediate values in sepsis, and the highest values in septic shock. Moreover, the sCD163 concentration was markedly higher in septic shock, significantly differing between conditions associated with lower (sepsis) and higher (septic shock) mortality. Interestingly, not only the values of the sCD163 concentration, but also the change of this concentration over the five days from ICU admission, are probably associated with in-hospital mortality. A five-day decrease in the sCD163 concentration was observed in survivors, whereas an increase in these values characterized non-survivors.

The research problem in many studies is to create predictive models supporting proper in-ward treatment. Pathological states associated with multiorgan failure (such as sepsis and septic shock) pose an exceptional problem in deriving these models, since many factors are involved in their pathomechanism. The two most simple types of predictive models shown in this study, logistic regression and Cox proportional hazards regression address, respectively, the odds and risk (hazard) of death among septic patients. According to the Cox proportional hazards regression model, IL-18 and sCD163 were significantly associated with the risk (hazard) of death among septic patients, even if one had the knowledge of the values of other parameters: APACHE II, WBC, PLT, PCT and CRP. Moreover, given that one did not use IL-18 and sCD163 for prediction, the process of risk assessment based solely on APACHE II, WBC, PLT, PCT and CRP would be relatively inferior to the model shown in this study (Table 6: model 1)—as assessed by the (pseudo-) R^2^ values (Table 6: model 1 vs. Table 6: model 2). Interestingly, the hazard ratios associated with sCD163 and IL-18 do not fit preliminary assumptions which would link increased IL-18 concentration with death—not survival (as shown in this study).

Patients with sepsis/septic shock undergo different disease stages. Both phases are dynamic and thus, during sepsis/septic shock, inflammation and immunosuppression may occur sequentially or concurrently. Sepsis may become fatal due to primary infection-induced excessive inflammation or immunosuppression. In a study of Volfovitch et al. assessing ferritin, sCD163 and IL-18 in COVID-19 patients, in contrast to our results, the levels of all analyzed factors were significantly higher in the non-survivors group. The authors [42] linked the increase in IL-18 and CD163 concentrations to two sepsis-associated phases: early pro-inflammatory and late, anti-inflammatory, respectively. Corroborating our findings, Feng et al. revealed that sCD163 was a useful marker activation of monocytes and macrophages and a promising prognostic marker of sepsis/septic shock mortality [43].

Taking into consideration the fact that ‘sepsis is associated with the dysregulation of the immunological system’, the study aimed to check whether predictive models could benefit from possessing the information on the values of serum concentrations of sCD163 and IL-18. The fact that, from a set of medical parameters (Table A3), sCD163 was iteratively chosen to be a part of the classification model (logistic regression) shows that the knowledge about the values of sCD163 concentration may potentially be used as an auxiliary feature in models predicting survival status, along with parameters which have been used in assessment of patients’ condition and sepsis severity (e.g., APACHE II and CRP, as featured in Table 4: model 2).

Based on the results, it could be hypothesized that in sepsis, although the pro-inflammatory part of the host response to the infection is markedly linked to the occurrence of multiorgan failure, the excessive upregulation of the anti-inflammatory part of this response is the factor which determines mortality rate. In this study there were some limitations which might have affected our results. Studies into this topic performed on a larger cohort and screening patients for longer periods of time might provide more information which could be used in the mortality risk assessment of septic patients.

## 5. Conclusions

Sepsis and septic shock are both accompanied with an immune response to the pathogens expressed as the activation of pro- and anti-inflammatory mediators. The dysregulated, extensive response of the immune system acts as one of the main causes of organ injury and high mortality rate in Intensive Care Units. The possibility of a dominating anti-inflammatory response in the early stages of sepsis/septic shock results in the priority of identifying patients who are likely to benefit from a targeted treatment modality. Early sCD163 monitoring might be included in the comprehensive strategy to evaluate the immune status regarding sepsis and septic shock and initiation of adequate therapy.

## Figures and Tables

**Figure 1 ijerph-20-02263-f001:**
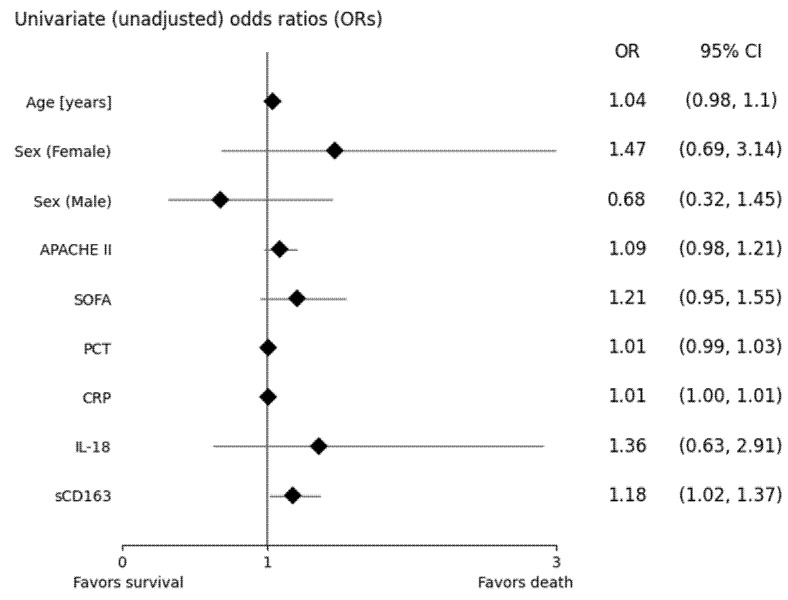
Forest plot of odds ratios (ORs) describing the odds of death among septic patients. ORs for IL-18 and sCD163 are given for log-transformed (base: 1.1.) values. APACHE II, acute physiology and chronic health evaluation II; SOFA, sequential organ failure assessment; PCT, procalcitonin; CRP, C-reactive protein; IL-18, interleukin 18; sCD163, soluble CD163.

**Figure 2 ijerph-20-02263-f002:**
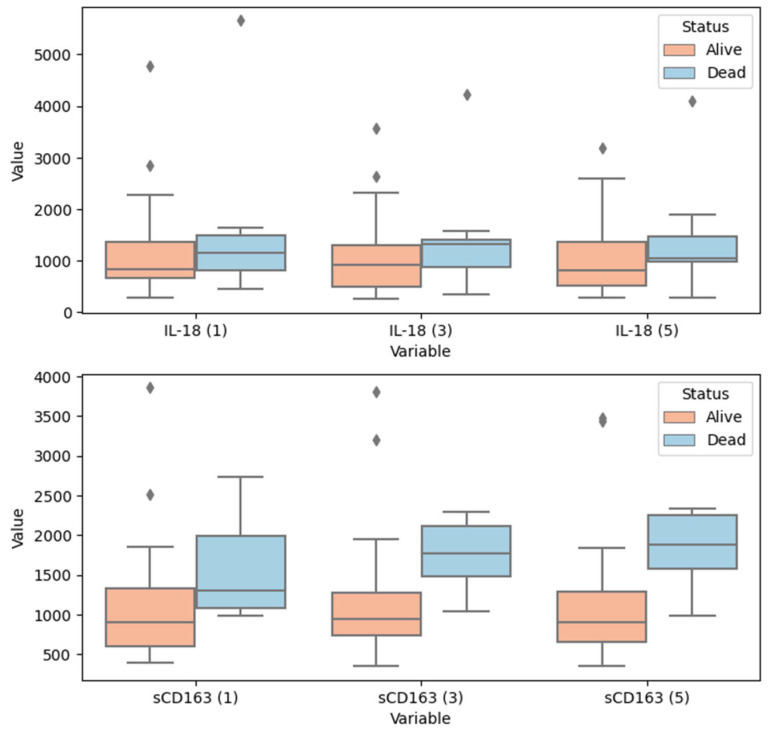
Boxplot of values of serum concentrations of IL-18 and sCD163 over five days (labelled in ‘()’ brackets in the variable names) since hospital admission in patients with diagnosed sepsis—in context of in-hospital survival. Fliers indicate outliers. IL-18, interleukin 18; sCD163, soluble CD163. Descriptive statistics and *p*-values associated with this plot are given in Table 5.

**Table 1 ijerph-20-02263-t001:** Clinical and demographic data describing the studied population sample.

Variable		Controls	Septic Patients	
Total Count (^c^)		30	40	
Diagnosis (^c^)		(Non-Applicable)	Sepsis	Septic Shock
			16	24
Age [years]	(^a^)	62.5 ± 8.57 {55; 61; 67}	65.5 ± 15.84 {58; 63; 78}	64.12 ± 13.13 {54.5; 65.5; 73}
	(^b^)			
APACHE II (upon admission)	(^a^)	-	20.75 ± 5.81 {16.5; 20.0; 26.0}	21.17 ± 7.81 {13.5; 22.0; 26.5}
	(^b^)			
SOFA (upon admission)	(^a^)	-	8.12 ± 3.55 {6.0; 7.0; 9.5}	9.46 ± 2.69 {7.5; 10.0; 11.0}
	(^b^)			
Female (^c^)		23	9	15
Male (^c^)		7	7	9
Source of infection: abdomen (^c^)		(non-specified)	8	22
Source of infection: lungs (^c^)			8	2
Survival status: survival (^c^)			16	13
Survival status: death (^c^)			0	11
Treatment: mechanical ventilation (^c^)			12	22
Treatment: catecholamines (^c^)			11	23
Organ failure: respiratory system (^c^)			13	23
Organ failure: cardiovascular system (^c^)			11	22
Organ failure: hematological system (^c^)			5	9
Organ failure: liver (^c^)			4	6
Organ failure: metabolic implications (^c^)			6	18
Organ failure: kidney(s) (^c^)			7	10
Organ failure: central nervous system (^c^)			10	13
Organ failure count: 2 (^c^)			4	3
Organ failure count: 3 (^c^)			8	4
Organ failure count: 4 (^c^)			0	7
Organ failure count: 5 (^c^)			1	6
Organ failure count: 6 (^c^)			2	3
Organ failure count: 7 (^c^)			1	1
Type of infection: bacterial Gram-negative (^c^)			8	12
Type of infection: bacterial, Gram-positive (^c^)			5	13
Type of infection: fungal (^c^)			2	4
Type of infection: unknown (^c^)			5	4
Mixed infection (2 or more pathogens) (^c^)			4	9

Presented values are expressed as mean value ± standard deviation (^a^) or {1st quartile; median value; 3rd quartile} (^b^) or actual number of patients (^c^). APACHE II, acute physiology and chronic health evaluation II; SOFA, sequential organ failure assessment.

**Table 2 ijerph-20-02263-t002:** First-day values of selected parameters in the three studied groups: control group, sepsis and septic shock.

Parameters	Control Group (C)	Sepsis (A)	Septic Shock (B)	*p* (C vs. A)	*p* (C vs. B)	*p* (A vs. B)	Global *p* (Ranked ANOVA)
IL-18 [pg/mL]	{126.24; **182.54**; 263.70}	{598.00; **735.50**; 877.00}	{849.50; **1243.00**; 1703.50}	**<0.0001**	**<0.0001**	0.3340	**<0.0001**
sCD163 [ng/mL]	{504.50; **661.90**; 777.10}	{509.95; **895.55**; 1230.85}	{945.20; **1240.15**; 1852.60}	0.2522	**<0.0001**	**0.0486**	**<0.0001**
sCD163/IL-18 ratio [ng/pg]	{2.67; **3.61**; 5.63}	{0.65; **0.99**; 1.64}	{0.55; **1.20**; 1.98}	<0.0001	**<0.000001**	**1.000**	**<0.0001**

Data are shown as {1st quartile; **median value (marked in bold)**; 3rd quartile}; *p*-values are **marked in bold** if <0.05. IL-18, interleukin 18; sCD163, soluble CD163.

**Table 3 ijerph-20-02263-t003:** Time-related changes in values of selected parameters in patients with sepsis and septic shock.

A: SEPSIS								
Parameters	1st Day (1)	3rd Day (3)	5th Day (5)	*p* (1 vs. 3)	*p* (1 vs. 5)	*p* (3 vs 5)	Global *p*	Sphericity *p*
SOFA [values]	8.13 ± 3.56	7.44 ± 3.97	6.63 ± 3.58	0.1494	**0.0007**	0.0752	**0.0010**	0.3107
Creatinine [mg/dL]	1.85 ± 0.96	1.69 ± 0.82	1.53 ± 0.67	0.3360	**0.0237**	0.3746	**0.0310**	0.4544
WBC [×10^3^/mm^3^]	13.27 ± 7.05	13.12 ± 5.26	13.24 ± 4.92	-	-	-	0.9464	**0.0005**
PCT [µg/L]	{2.27; **5.00**; 16.01}	{1.57; **3.51**; 11.30}	{0.95; **2.23**; 6.84}	0.3400	0.1000	**<0.010**	**0.0010**	-
CRP [mg/L]	170.08 ± 100.93	157.20 ± 88.34	114.92 ± 56.96	0.6429	**0.0024**	**0.0186**	**0.0334**	**0.0290**
IL-18 [pg/mL]	{598.00; **735.50**; 877.00}	{448.00; **600.00**; 944.50}	{444.00; **554.00**; 1191.50}	-	-	-	0.0874	-
sCD163 [ng/mL]	915.91 ± 429.14	937.51 ± 457.22	965.19 ± 491.24	-	-	-	0.8591	0.9793
sCD163/IL-18 ratio [ng/pg]	1.24 ± 0.80	1.66 ± 1.45	1.60 ± 1.41	-	-	-	0.2321	**0.0117**
**B: SEPTIC SHOCK**								
**Parameters**	**1st Day (1)**	**3rd Day (3)**	**5th Day (5)**	***p* (1 vs. 3)**	***p* (1 vs. 5)**	***p* (3 vs 5)**	**Global *p***	**Sphericity *p***
SOFA [values]	{7.50; **10.00**; 11.00}	{7.50; **9.50**; 12.00}	{7.00; **10.00**; 12.00}	-	-	-	0.8698	-
Creatinine [mg/dL]	{0.98; **1.28**; 1.65}	{0.78; **1.12**; 1.46}	{0.69; **0.97**; 1.42}	0.93	**<0.01**	**0.035**	**0.0012**	-
WBC [×10^3^/mm^3^]	{11.35; **15.41**; 22.09}	{11.63; **12.72**; 24.26}	{8.54; **12.68**; 19.06}	-	-	-	0.4378	-
PCT [µg/L]	{4.93; **25.82**; 43.15}	{7.97; **20.93**; 44.28}	{7.51; **15.91**; 30.48}	1	**<0.01**	0.09	**0.0076**	-
CRP [mg/L]	{192.15; **235.48**; 257.65}	{125.30; **200.72**; 226.14}	{114.00; **163.00**; 219.00}	0.19	**0.02**	**<0.01**	**<0.0001**	-
IL-18 [pg/mL]	{849.50; **1243.00**; 1703.50}	{881.00; **1241.50**; 1636.00}	{795.50; **1081.50**; 1521.50}	-	-	-	0.0787	-
sCD163 [ng/mL]	{945.20; **1240.15**; 1852.60}	{967.65; **1352.30**; 1993.65}	{904.30; **1361.60**; 2163.30}	-	-	-	0.4169	-
sCD163/IL-18 ratio [ng/pg]	1.26 ± 0.73	1.40 ± 0.96	1.59 ± 1.20	-	-	-	0.1088	**<0.0001**

Data are shown as mean value ± standard deviation or {1st quartile; **median value (marked in bold)**; 3rd quartile}; *p*-values are **marked in bold** if <0.05. “Global *p*” shows, by default, *p*-values for the Friedman test (if median values are analysed) or the F test for repeated measures of the ANOVA (if mean values are analysed). “Sphericity *p*” shows *p*-values for Mauchly’s test. When there was no sphericity (*p* < 0.05), multivariate analysis (Pillai’s trace) was performed (also shown in “global *p*”). SOFA, sequential organ failure assessment; WBC, white blood cell count; PCT, procalcitonin; CRP, C-reactive protein; IL-18, interleukin 18; Scd163, Soluble CD163.

**Table 4 ijerph-20-02263-t004:** Logistic regression models of the best fit describing the association between the selected predictors and in-hospital death in septic patients.

**PREDICTION MODEL 1 (Null Hypothesis Score Test *p* = 0.0064)** **In the process of deriving this model, sCD163 and IL-18 were used as separate features**						
$P\;\left(Y=“death”x_{1},\;x_{2}\right)=\frac{e^{\left(-27.345+0.087x_{1}+0.277x_{2}\right)}}{1+e^{\left(-27.345+0.087x_{1}+0.277x_{2}\right)}}$						
**AIC**	**BIC**	**Pseudo-R^2^ (Nagelkerke)**	**Hosmer-Lemeshow Test *p***		**AUC (Learning)**	**AUC (Testing)**
40.82	45.88	0.3811	0.6811		0.828 ± 0.0655	0.792 ± 0.0709
**Variable**		**β_i_**	**OR**	**OR CI (−95%)**	**OR CI (95%)**	** *p* **
y intercept (β_0_)		−27.345	-	-	-	-
$\left[x_{1}\right]$ Age		0.087	1.091	1.009	1.179	0.029
$\left[x_{2}\right]$ log_1.1_[sCD163]		0.277	1.319	1.075	1.618	0.008
**PREDICTION MODEL 2 (Null Hypothesis Score Test *p* = 0.0220)** **In the process of deriving this model, the sCD163/IL-18 ratio was used instead of its components: sCD163 and IL-18**						
$P\;\left(Y=“death”x_{1},\;x_{2},x_{3}\right)=\frac{e^{\left(-8.877+1.361x_{1}+0.009x_{2}+0.174x_{3}\right)}}{1+e^{\left(-8.877+1.361x_{1}\pm 0.009x_{2}+0.174x_{3}\right)}}$						
**AIC**	**BIC**	**Pseudo-R^2^ (Nagelkerke)**	**Hosmer-Lemeshow Test *p***		**AUC (Learning)**	**AUC (Testing)**
43.54	50.29	0.3617	0.3228		0.821 ± 0.0733	0.729 ± 0.0899
**Variable**		**β_i_**	**OR**	**OR CI (−95%)**	**OR CI (95%)**	** *p* **
y intercept (β_0_)		−8.877	-	-	-	-
$\left[x_{1}\right]$ sCD163/IL-18		1.361	3.900	1.015	14.989	0.048
$\left[x_{2}\right]$ CRP		0.009	1.009	1.001	1.018	0.026
$\left[x_{3}\right]$ APACHE II		0.174	1.190	1.023	1.384	0.024

Admission values were used by the model. AIC, Akaike Information Criterion; BIC, Bayesian Information Criterion; AUC, area under the curve; sCD163, soluble CD163. The stepwise process of model derivation is shown in Table A3.

**Table 5 ijerph-20-02263-t005:** Differences, among septic patients, in values of IL-18 and sCD163, in context of survival status and analysed timepoint (day of hospitalization).

Parameters (Timepoints)	Survival Status: Alive	Survival Status: Dead	*p*
IL-18 [pg/mL] (1st day)	{671.00; **846.00**; 1372.00}	{762.00; **1147.00**; 1629.00}	0.455
IL-18 [pg/mL] (3rd day)	{504.00; **926.00**; 1296.00}	{804.00; **1328.00**; 1420.00}	0.254
IL-18 [pg/mL] (5th day)	{528.00; **818.00**; 1364.00}	{985.00; **1037.00**; 1611.00}	0.241
sCD163 [ng/mL] (1st day)	{596.90; **906.50**; 1323.20}	{1074.10; **1294.20**; 2114.10}	**0.017**
sCD163 [ng/mL] (3rd day)	{736.80; **942.30**; 1276.80}	{1386.10; **1763.60**; 2176.20}	**0.0005**
sCD163 [ng/mL] (5th day)	{653.70; **898.80**; 1280.30}	{1546.20; **1880.80**; 2307.00}	**0.0001**

Data are shown as {1st quartile; **median value (marked in bold)**; 3rd quartile}; *p*-values are **marked in bold** if <0.05. IL-18, interleukin 18; sCD163, soluble CD163.

**Table 6 ijerph-20-02263-t006:** Cox proportional hazards regression models.

MODEL 1 (Model of the Best Fit; Iterative Process: Inclusion if *p* < 0.05; Exclusion if *p* > 0.10)							
**−2LogL**		**AIC**		**BIC**		**R^2^**	
46.84		60.84		63.62		0.7600	
**Variable**	**β_i_**	**β_i_ SE**	**Χ^2^**	** *p* **	**HR**	**HR −95% CI**	**HR 95% CI**
APACHE II	0.1980	0.0804	6.0569	0.0139	1.2189	1.0411	1.4271
WBC	0.0757	0.0429	3.1135	0.0776	1.0786	0.9917	1.1732
PLT	0.0074	0.0046	2.5453	0.1106	1.0074	0.9983	1.0166
PCT	0.0211	0.0119	3.1606	0.0754	1.0214	0.9978	1.0455
CRP	0.0073	0.0037	3.9736	0.0462	1.0073	1.0001	1.0145
IL-18	−0.0009	0.0004	4.7997	0.0285	0.9991	0.9982	0.9999
sCD163	0.0018	0.0008	5.0452	0.0247	1.0018	1.0002	1.0033
**MODEL 2 (Alternative Model—Model 1 without the Information on CD163 and IL-18)**							
**−2LogL**		**AIC**		**BIC**		**R^2^**	
55.13		65.13		67.12		0.4831	
**Variable**	**β_i_**	**β_i_ SE**	**Χ^2^**	** *p* **	**HR**	**HR −95% CI**	**HR 95% CI**
APACHE II	0.0837	0.0553	2.2909	0.1301	1.0873	0.9756	1.2118
WBC	0.0429	0.0319	1.8051	0.1791	1.0438	0.9805	1.1112
PLT	0.0028	0.0036	0.5988	0.4391	1.0028	0.9957	1.0099
PCT	0.0124	0.0103	1.4652	0.2261	1.0125	0.9923	1.0331
CRP	0.0042	0.0030	1.9091	0.1671	1.0042	0.9982	1.0102

Both models used information gained upon admission. β_i_, regression coefficient; SE, standard error; logL, log(likelihood); AIC, Akaike Information Criterion; BIC, Bayesian Information Criterion; HR, hazard ratio; CI, confidence interval; APACHE II, acute physiology and chronic health evaluation II; WBC, white blood cell count; PLT, blood platelet count; PCT, procalcitonin, CRP, C-reactive protein; IL-18, interleukin 18; sCD163, soluble CD163.

## Data Availability

The data are available from the corresponding author upon reasonable request.

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
