# Peer review of "Alterations in Serum Concentration of Soluble CD163 within Five Study Days from ICU Admission Are Associated with In-Hospital Mortality of Septic Patients—A Preliminary Study"

_ijerph, 2023, doi:10.3390/ijerph20032263_

Round 1

Reviewer 1 Report

It is an interesting study that brings information with immediate practical applicability. The manuscript is very well written and structured, with some minor observations I will detail in the following. However, I consider having a small number of patients studied, and last but not least, the importance of the theme addressed by the authors, that the paper should be classified as a short communication.

In conclusion, I propose the manuscript for publication with the decision of minor revisions.

2. Materials and methods

2.1. Study populations

Lines 88-92: I consider the number of patients selected for the study small. During what period did the study take place in the hospital unit? Was Sampling Bias taken into account?

Line 182: Table 1. Clinical and demographic data describing the studied population sample

I recommend redoing the table so that the information provided is easy to follow and understand. In the first part, there are some numbers in brackets, but we don't know what they refer to, and in the second part, there is no continuity; there is a certain value. If somehow the figures represent the mean and the standard deviation, then this must be highlighted.

Author Response

The Authors are thankful for all the remarks. The manuscript has been revised along with the provided suggestions. Please, find the detailed response below.

It is an interesting study that brings information with immediate practical applicability. The manuscript is very well written and structured, with some minor observations I will detail in the following. However, I consider having a small number of patients studied, and last but not least, the importance of the theme addressed by the authors, that the paper should be classified as a short communication.

In conclusion, I propose the manuscript for publication with the decision of minor revisions.

Thank You kindly for this decision. We believe that the quality of the revised manuscript has been improved.

  1. Materials and methods

2.1. Study populations

Lines 88-92: I consider the number of patients selected for the study small. During what period did the study take place in the hospital unit? Was Sampling Bias taken into account?

The number of patients in our study was small due to low financial possibilities. We collected samples from the patients who were admitted to one Intensive Care Unit. We had to choose the most optimal strategy for study design among two possible: 1) to increase the length of the study and reduce the count of participants or 2) to reduce the length of the study to first five days from admission and keep the current count of participants. We chose the latter as a preliminary character of the study was pre-assumed.

Our study was observational and prospective (an excerpt on this matter has been added to the ‘Materials and Methods’ section. Our study took place between the period November 2016 and November 2017. This information has, likewise, been added.

Line 182: Table 1. Clinical and demographic data describing the studied population sample

I recommend redoing the table so that the information provided is easy to follow and understand. In the first part, there are some numbers in brackets, but we don't know what they refer to, and in the second part, there is no continuity; there is a certain value. If somehow the figures represent the mean and the standard deviation, then this must be highlighted.

Thank You for this suggestion. Additional marks with description of the featured values have been added to Table 1. Hopefully, this form would be more transparent for the potential Readers.

Reviewer 2 Report

In this preliminary study, sCD163 and IL-18 were invesigated as potential biomarkers for the prognosis of mortality in sepsis and septic shock. It was shown that these biomarkers combined with other markers, clinical scores and demographic data improve the prediction of mortality in ICU patients.

The paper is well written and the results are nicely presented however, the results cannot provide very accurate conclusions. A different data analysis of the data could improve this. Kindly find my comments below.

Table 4: It is not clear to me from the text (methods and results section) which parameters were included into the model. Are the selected predictors all parameters mentioned in Figure 1 or just CD18 and sCD163? Kindly specify in detail in the text.

Also, why perform 10-fold CV? Patient-out CV is more accurate.

Figure 2/Lines 341-345: Kindly mention statistical significance between dead/alive patients in the plot, especially in the case of sCD163 where differences are mentioned to be observed.

Lines 257-261: Kindly mention what parameter you are referring to, ROC of what?

Lines 283-284/363-365: These statements are not very accurate since pro- and anti-inflammatory response is happen simultaneously aiming the immune response regulation, as also mentioned by the authors (lines 285-286). Maybe these sentences should be removed to avoid confusion.

Lines 331-336: Since sepsis is defined as a hyperactivation of immune response how do you explain that higher IL-18 levels result in decrease in hazard rate? The opposite should be expected.  

It would be a surprise if the predictive values in terms of sensitivity and specificity would be accurate with only 40 patients enrolled. This, most likely, is also the reason for no statistical difference observed in the time-related changes. The differences between patients were too high to provide significant results, considering the high variance in the source of infection, severity and pathogen types, as mentioned in Table 1.  Thus, no specific conclusions can be drawn from these data and this analysis on the positive or negative effect of sCD163 and IL-8 on survival. However, since one molecule is pro- and the other anti-inflammatory, maybe the ratio of their serum levels could provide better inside into the patient’s immune status. Have you tried this analysis? According on your data, the survivors have a more inflammatory profile (higher IL-18 and lower CD163) and the non-survivors a less inflammatory response. When looking at the data from this point of view maybe better conclusions can be drawn regarding the risk of death.

Author Response

The Authors are thankful for all the remarks. The manuscript has been revised along with the provided suggestions. Please, find the detailed response below.

In this preliminary study, sCD163 and IL-18 were investigated as potential biomarkers for the prognosis of mortality in sepsis and septic shock. It was shown that these biomarkers combined with other markers, clinical scores and demographic data improve the prediction of mortality in ICU patients.

The paper is well written and the results are nicely presented however, the results cannot provide very accurate conclusions. A different data analysis of the data could improve this. Kindly find my comments below.

Thank You for this commentary. Due to Your suggestions, the manuscript has been vastly improved and new insights on the data were gained.

Table 4: It is not clear to me from the text (methods and results section) which parameters were included into the model. Are the selected predictors all parameters mentioned in Figure 1 or just CD18 and sCD163? Kindly specify in detail in the text.

The process of model derivation could be followed, step-by-step, in Table A3. This information was, previously, given in the materials and method section. In the revised version of the manuscript, this information is also included in the footer of the table in which the models are presented. Thank You kindly for this suggestion. The manuscript should be more transparent now.

Also, why perform 10-fold CV? Patient-out CV is more accurate.

Thank You for this remark. It is true that leave-one-out CV (LOOCV) could be more precise given that sample size was small. However, LOOCV is not supported by the Statistica package which we used for deriving the models. Moreover, obtaining very accurate metrics in a small population sample is unfeasible, regardless of the used CV method. The study was aimed to explore the possible predictors in order to set some foundations for future experiments performed on a vastly larger cohort. We have added information on the lack of use of LOOCV in the “Statistical analysis” section. A new paragraph which mentions the concept of the use of AI for prediction has been added to provide background for the commentary on the merits of using LOOCV instead of 10-fold CV.

Figure 2/Lines 341-345: Kindly mention statistical significance between dead/alive patients in the plot, especially in the case of sCD163 where differences are mentioned to be observed.

Thank You for mentioning the lack of p-values. Formerly, we wanted to show only the association between time and the reported contrasts between these two groups – to potentially provide an idea to feature this difference in predictive models. Based on Your remark, we have revised the manuscript, introducing a table with descriptive statistics and p-values.

Lines 257-261: Kindly mention what parameter you are referring to, ROC of what?

Table A4 has been updated with model labels. P(Y) is the score calculated with formulae given in Table 4. This information is now given in table footer. Thank You for pointing this out.

Lines 283-284/363-365: These statements are not very accurate since pro- and anti-inflammatory response is happen simultaneously aiming the immune response regulation, as also mentioned by the authors (lines 285-286). Maybe these sentences should be removed to avoid confusion.

It is true that the greatly convoluted form of these sentences could confuse the potential Readers. The sentences have been re-written as (we believe) more accessible for potential Readers.

Lines 331-336: Since sepsis is defined as a hyperactivation of immune response how do you explain that higher IL-18 levels result in decrease in hazard rate? The opposite should be expected.

This is an interesting question. Hazard rate describes a very specific type of risk adjusted with the time course of the study. The study described in this manuscript explored the initial phase of sepsis. Based on the fact that, at each studied timepoint, the non-survivors showed significantly higher sCD163 concentration compared to the survivors, it could be hypothesized that, at the onset of sepsis, the organism maintains the capacity to modulate the pro-inflammatory response by liberating sCD163 (and, probably, other anti-inflammatory agents) into the extracellular fluid. It could be assumed that the risk of death is related not with sCD163 or IL-18 per se, but - with a lack of pro-/anti-inflammatory homeostasis.

We believe that the fact that IL-18 was negatively affecting the hazard stems from the positive relation between sCD163 and the hazard, and the antagonistic nature of IL-18 and CD163. The hazard ratio associated with IL-18 was very little (about 0.09%). Moreover, the hazard ratio of IL-18 was estimated under the presence of various inflammation-associated features (such as APACHE II, WBC and CRP) which were more important in the model (based on their βi coefficients). Thus, both CD163 and IL-18 could be tested as auxiliary features for estimating the risk of death. This insight has been added to the discussion section.

Your remark gave us food for thought as, presumably given more time, the inflammatory response would remain to be quenched to a point in which the organism would fail to adapt to the increase in pro-inflammatory response. Determining the time in which this lack of anti-inflammatory response would happen in septic patients in a study longer than the one presented in this manuscript would be valuable for risk assessment as, perhaps, the association between sCD163 and IL-18 might not be linear (given that the response of the organism to various stimuli is dynamic)? We will try to answer this question in future studies. Thank You kindly for inviting us to discuss this topic.

It would be a surprise if the predictive values in terms of sensitivity and specificity would be accurate with only 40 patients enrolled. This, most likely, is also the reason for no statistical difference observed in the time-related changes. The differences between patients were too high to provide significant results, considering the high variance in the source of infection, severity and pathogen types, as mentioned in Table 1.  Thus, no specific conclusions can be drawn from these data and this analysis on the positive or negative effect of sCD163 and IL-8 on survival. However, since one molecule is pro- and the other anti-inflammatory, maybe the ratio of their serum levels could provide better inside into the patient’s immune status. Have you tried this analysis? According on your data, the survivors have a more inflammatory profile (higher IL-18 and lower CD163) and the non-survivors a less inflammatory response. When looking at the data from this point of view maybe better conclusions can be drawn regarding the risk of death.

This is a valid point. Indeed, the study may only provide the foundation for future investigations – thus, throughout the text, the study is termed as ‘preliminary’. By no means are the metrics (precision, specificity etc.) objectively accurate. The discussion section has been updated with the information on this lack of accuracy. Hopefully, future studies would provide more accurate metrics.

Tables 2 were expanded with information on the sCD163/IL-18 ratio and its respective group-wise or time-related differences.

Based on Your remark we have also come up with an idea of putting the IL-18/CD163 ratio among the variables initially taken into consideration in the process of logistic regression model derivation. We have followed this approach, by featuring the sCD163/IL-18 ratio (instead of separate variables: sCD163 and IL-18) among the initial features used in iteration. Table 2 has been expanded with this newly derived model. In this model, three features are used: the CD163/IL-18 ratio (raw ratio value since it is linear vs. log(odds), unlike raw sCD163 values), CRP concentration and APACHE II values. The appendix has been expanded with the thorough information on the stepwise process of model derivation and he metrics of the model, along with the Youden’s J index. Thank You kindly for suggesting this approach.

Round 2

Reviewer 2 Report

The authors provided very detailed and presice answere to the reviewers questions. The suggestions were incorposrated into the manuscript in a very detailed manner that was beyond my expectations. The manuscript has now significanly impoove and is ready for publication. 

I would like to thank the authors for their kind collaboration in this peer-review process.